



**The effect of effective rock viscosity on 2D magmatic porosity waves**

By

Janik Dohmen, Harro Schmeling and Jan Philipp Kruse

Institute for Geoscience, Goethe University Frankfurt, Germany

10                    Submitted to Solid Earth (EGU)

26.6.2019



**Abstract**

In source regions of magmatic systems the temperature is above solidus and melt ascent is assumed to

occur predominantly by two-phase flow which includes a fluid phase (melt) and a porous deformable

matrix. Since McKenzie (1984) introduced his equations for two-phase flow, numerous solutions have

been studied one of which predicts the emergence of solitary porosity waves. By now most analytical

and numerical solutions for these waves used strongly simplified models for the shear- and bulk viscosity

of the matrix, significantly overestimating the viscosity or completely neglecting the porosity-

dependence of the bulk viscosity. Schmeling et al. (2012) suggested viscosity laws in which the viscosity

decreases very rapidly for small melt fractions. They are incorporated into a 2D finite difference mantle

convection code with two-phase flow (FDCON) to study the ascent of solitary porosity waves. The

models show that, starting with a Gaussian shaped wave, they rapidly evolve into a solitary wave with

similar shape and a certain amplitude. Despite the strongly weaker rheologies compared to previous

viscosity laws the effect on dispersion curves and wave shape are only moderate as long as the

background porosity is fairly small. The models are still in good agreement with semi-analytic solutions

which neglect the shear stress term in the melt segregation equation. However, for higher background

porosities and wave amplitudes associated with a viscosity decrease of 50% or more, the phase velocity

and the width of the waves are significantly decreased. Our models show that melt ascent by solitary

waves is still a viable mechanism even for more realistic matrix viscosities.

**1. Introduction**

Magmatic phenomena such as volcanic eruptions on the earth's surface show, among others, that melt is

able to ascend from partially molten regions in the earth's mantle. The melt initially segregates through

the partially molten source region and then ascends through the unmolten lithosphere until it eventually

reaches the surface. Within supersolidus source regions at low melt fractions melt is assumed to slowly

percolate by two-phase porous flow within a deforming matrix (McKenzie, 1984; Schmeling, 2000;

Bercovici et al., 2001), followed by melt accumulation within rising high porosity waves (Scott and

Stevenson, 1984; Spiegelman, 1993, Wiggins and Spiegelman, 1995; Richard et al., 2012) or focusing into

channels (Stevenson, 1989; Richardson, 1998) which have the potential to penetrate into subsolidus

regions above to generate dykes. Here we focus on the supersolidus source region, and in particular on

the dynamics of porosity waves. An essential parameter controlling the width and ascent velocity of

porosity waves is the effective shear and bulk matrix viscosity (Simpson and Spiegelman, 2011; Richard





et al., 2012). So far most of the porosity wave model approaches used either equal bulk and shear

viscosities, or simple laws in the form of

$$\eta_s = \eta_{s0}(1 - \varphi), \tag{1}$$

$$\eta_b = \eta_{s0} C \frac{(1-\varphi)}{\varphi} \tag{2}$$

where $\eta_s$ is the effective shear viscosity of the matrix, $\eta_b$ the bulk viscosity, $\eta_{s0}$ the intrinsic shear

viscosity of the matrix, $C$ a constant of order 1, and $\varphi$ the porosity. Schmeling et al. (2012) developed an

effective viscosity model depending on a simplified geometry of the fluid phase within a viscous matrix.

Possible melt geometries include flat, ellipsoid-shaped melt inclusions with an aspect ratio $\alpha$ and melt

tubes with circular or triangular cross sections with tapered edges. Comparison of the previous (1), (2)

viscosity laws with the ones by Schmeling et al. (2012) clearly shows that for aspect ratio 1 and

particularly for smaller $\alpha$ the effective matrix viscosities are significantly weaker, and disaggregation of

the solid occurs at melt fractions significantly smaller than 100% as predicted by laws (1) and (2). Recent

viscosity models based on microscopic diffusion through grains, grain faces and the melt phase confirm

the significance of weakening with respect to equations (1) and (2) (Rudge, 2018). The aim of this study is

to model 2D-porosity waves with the viscosity laws by Schmeling et al. (2012) and test the influence of

the weaker rheology on their shape and ascent velocity in the absence of melting or freezing.

**2. Theoretical Approach**

**2.1 Governing equations**

The mathematical formulation of differential movement between solid matrix and melt basically builds

on that described in Schmeling (2000) and Schmeling et al. (2019) and is applied here to a porosity wave.

We solve the equations for mass and momentum conservation for melt and solid. The formulation of the

governing equations for the melt-in-solid two-phase flow dynamics is based on McKenzie (1984),

Spiegelman & McKenzie (1987) and Schmeling (2000) and is valid for infinite Prandtl number (i.e.

neglecting inertia terms in the momentum equations), and small fluid to matrix viscosity ratios. In the

following all variables associated with the fluid (melt) have the subscript $f$ and those associated with the

solid have the subscript $s$. Without melting and freezing the equation for the conservation of the mass of

the melt is

$$\rho_f \left( \frac{\partial \varphi}{\partial t} + \vec{\nabla} \cdot (\varphi \vec{v}_f) \right) = 0, \tag{3}$$



and the mass conservation of the solid is

$$\rho_s \left( \frac{\partial(1-\varphi)}{\partial t} + \vec{\nabla} \cdot ((1-\varphi)\vec{v}_s) \right) = 0. \tag{4}$$

$\rho_f$ and $\rho_s$ are the melt and solid densities, which are assumed constant in eqs. (3) and (4) but different, $\vec{v}_f$ and $\vec{v}_s$ are the fluid and solid velocities, respectively. The velocities are derived from the momentum equations, which is a generalized Darcy equation for the fluid separation flow

$$\vec{v}_f - \vec{v}_s = -\frac{k_\varphi}{\eta_f\, \varphi}\left(\vec{\nabla}P - \rho_f \vec{g}\right), \tag{5}$$

and the Stokes equation for the solid-fluid mixture in the limit of zero fluid viscosity

$$\rho \vec{g} - \vec{\nabla}P + \frac{\partial \tau_{ij}}{\partial x_j} = 0. \tag{6}$$

$k_\varphi$ is the permeability that depends on the rock porosity (i.e. melt fraction) with the power $n$

$$k_\varphi = k_0 \varphi^n, \tag{7}$$

$\eta_f$ is the dynamic melt viscosity, $\vec{g}$ is the gravitational acceleration, $\rho$ is the density of the melt – solid mixture, $P$ is the pressure, whose gradient is driving the motion, and $\tau_{ij}$ is the viscous stress tensor

$$\tau_{ij} = \eta_s \left( \frac{\partial v_{si}}{\partial x_j} + \frac{\partial v_{sj}}{\partial x_i} \right) + \left( \eta_b - \frac{2}{3}\eta_s \right) \delta_{ij} \vec{\nabla} \cdot \vec{v}_s \tag{8}$$

with the effective shear viscosity $\eta_s$ and the effective volumetric or bulk viscosity $\eta_b$ of the porous matrix. The term $\left( \eta_b - \frac{2}{3}\eta_s \right) \vec{\nabla} \cdot \vec{v}_s$ is often referred to as compaction pressure. The linearized equations of state for the mixture density and the fluid density are given as

$$\rho = \rho_0 \left(1 - c_f \varphi\right) \tag{9}$$

and

$$\rho_f = \rho_0 \left(1 - c_f\right), \tag{10}$$

respectively, and $c_f = \frac{\rho_0 - \rho_{0f}}{\rho_0}$.



Neglecting capillary pressure at the melt-solid interfaces, the pressure is equal in the melt phase and in the solid phase. With this assumption eq. (5) and eq. (6) can be merged to eliminate the pressure. Inserting the densities of the mixture and the fluid, and using eq. (7), eq. (5) is recast into


$$\vec{v}_f - \vec{v}_s = -\frac{k_0 \varphi^{n-1}}{\eta_f}\left(\rho_0 \vec{g} c_f (1-\varphi) + \frac{\partial \tau_{ij}}{\partial x_j}\right). \tag{11}$$

This equation states that the velocity difference between fluid and solid phase (i.e. fluid separation flow, or the segregation velocity) is driven by the buoyancy of the fluid with respect to the solid, and the viscous stress in the matrix which includes the compaction pressure.

Following Šrámek *et al.* (2007) the matrix velocity, $\vec{v}_s$, can be written as the sum of the incompressible
flow velocity, $\vec{v}_1$, and the irrotational (compaction) flow velocity, $\vec{v}_2$, as follows:

$$\vec{v}_s = \vec{v}_1 + \vec{v}_2 = \begin{pmatrix}\frac{\partial \psi}{\partial z} \\ -\frac{\partial \psi}{\partial x}\end{pmatrix} + \begin{pmatrix}\frac{\partial \chi}{\partial x} \\ \frac{\partial \chi}{\partial z}\end{pmatrix} \tag{12}$$

with the incompressible velocity potential or stream function $\psi$ and the irrotational (compaction related) velocity potential, $\chi$. From eq. (12) it follows that the latter is given as the solution of the Poisson equation


$$\nabla^2 \chi = \vec{\nabla} \cdot \vec{v}_s \tag{13}$$

The divergence term $\vec{\nabla} \cdot \vec{v}_s$ can be derived from summing up eq. (3) and eq. (4) to give

$$\vec{\nabla} \cdot \vec{v}_s = -\vec{\nabla} \cdot \left[\varphi\left(\vec{v}_f - \vec{v}_s\right)\right] \tag{14}$$

Eq. (13) represents a Poisson equation which can be solved for $\chi$ once the melt porosity and segregation velocity are known. As boundary condition the normal velocity of $\vec{v}_2$, i.e. $v_{2n}$, can be prescribed which is
equivalent to a normal derivative of $\chi$, i.e. a Neuman boundary condition. If the normal velocity is constant along the boundary, it automatically fulfills free slip. For sake of simplicity $v_{2n} = 0$ was chosen.

Taking the curl of the matrix momentum eq. (6) eliminates the pressure. Inserting the viscous stress tensor (eq. 8), the density (eq. 9) and the matrix velocity (eq. 12) into the resulting equation gives the momentum equation in terms of the stream function $\psi$ and the irrotational velocity potential $\chi$


$$\left(\frac{\partial^2}{\partial x^2} - \frac{\partial^2}{\partial z^2}\right)\left[\eta_s\left(\frac{\partial^2 \psi}{\partial x^2} - \frac{\partial^2 \psi}{\partial z^2}\right)\right] + 4\frac{\partial^2}{\partial x \partial z}\left[\eta_s \frac{\partial^2 \psi}{\partial x \partial z}\right] = (\rho_s - \rho_f)g\frac{\partial \varphi}{\partial x} + A(\chi) \tag{15}$$

with



$$A(\chi) = -2\frac{\partial^2}{\partial x \partial z}\left[\eta_s\left(\frac{\partial^2 \chi}{\partial x^2} - \frac{\partial^2 \chi}{\partial z^2}\right)\right] + 2\left(\frac{\partial^2}{\partial x^2} - \frac{\partial^2}{\partial z^2}\right)\left[\eta_s\frac{\partial^2 \chi}{\partial x \partial z}\right]$$

The governing equations are non-dimensionalized by the compaction length (McKenzie, 1984) and a scaling separation velocity. The compaction length is given by

$$\delta_c = \left(\frac{\eta_b + \frac{4}{3}\eta_s}{\eta_f}k_\varphi\right)^{\frac{1}{2}} \tag{16}$$

and is the length scale over which a variation in fluid flux gives a response on the compaction. The scaling separation velocity is given as

$$v_{sc} = \frac{k_\varphi}{\eta_f \varphi}(\rho_0 - \rho_{0f})g \tag{17}$$

Both scaling quantities are taken at a reference state which assumes a constant background porosity $\varphi_0$. This defines the scaling law, where the primes denote non-dimensional values and the subscript 0 refers to the background porosity

$$\vec{x} = \delta_{c0}\,\vec{x}' \qquad \vec{v} = v_{sc0}\vec{v}' \qquad t = \frac{\delta_{c0}}{v_{sc0}}t' \qquad \tau_{ij} = \eta_{s0}\frac{v_{sc0}}{\delta_{c0}}\tau_{ij}' \tag{18}$$

$$\eta = \eta_{s0}\eta' \qquad \rho = \rho_{s0}\rho' \qquad \varphi = \varphi_0\varphi'$$

Dropping the constant densities, the resulting governing equations for the mass are

$$\frac{\partial(1-\varphi')}{\partial t'} + \vec{\nabla}' \cdot \left((1-\varphi')\vec{v}_s'\right) = 0 \tag{19}$$

$$\frac{\partial \varphi'}{\partial t'} + \vec{\nabla}' \cdot \left(\varphi'\vec{v}_f'\right) = 0 \tag{20}$$

and for the momentum equations we get

$$\left(\frac{\partial^2}{\partial x'^2} - \frac{\partial^2}{\partial z'^2}\right)\left[\eta_s'\left(\frac{\partial^2 \psi'}{\partial x'^2} - \frac{\partial^2 \psi'}{\partial z'^2}\right)\right] + 4\frac{\partial^2}{\partial x'\partial z'}\left[\eta_s'\frac{\partial^2 \psi'}{\partial x'\partial z'}\right] = \varphi_0^2\frac{\eta_{b0}+\frac{4}{3}\eta_{s0}}{\eta_{s0}}\frac{\partial \varphi'}{\partial x'} + A(\chi') \tag{21}$$

$$A(\chi') = -2\frac{\partial^2}{\partial x'\partial z'}\left[\eta_s'\left(\frac{\partial^2 \chi'}{\partial x'^2} - \frac{\partial^2 \chi'}{\partial z'^2}\right)\right] + 2\left(\frac{\partial^2}{\partial x'^2} - \frac{\partial^2}{\partial x'^2}\right)\left[\eta_s'\frac{\partial^2 \chi'}{\partial x'\partial z'}\right]$$

$$\vec{v}_f' - \vec{v}_s' = \varphi'^{n-1}\left((1-\varphi_0\varphi')\vec{e}_z - \frac{\eta_{s0}}{\left(\eta_{b0}+\frac{4}{3}\eta_{s0}\right)}\frac{1}{\varphi_0}\frac{\partial \tau_{ij}'}{\partial x_{j}'}\right) \tag{22}$$





with $\vec{e}_z$ as unit vector in z- direction (positive upward).

## 2.2. The effective viscosity of a porous matrix

The effective viscosity laws proposed by Schmeling et al. (2012) assume ellipsoidal melt inclusions, or
melt films if the inclusions are flat, or melt tubules embedded within an effective viscous medium. This self-consistent assumption is able to predict viscous weakening of a solid matrix with a disaggregation melt porosity of the order of 50% or less depending on the assumed melt geometry. From their numerical models Schmeling et al. (2012) derive approximate formulas for the porosity dependence of the effective matrix shear and bulk viscosities for a melt network geometry consisting of 100% films


$$\eta_s = \eta_{s0}\left(1 - \frac{\varphi}{c_1}\right)^{k_1} \quad \text{for } \varphi < c_1 \tag{23}$$

$$\eta_b = \eta_{s0}c_2\frac{(c_1-\varphi)^{k_2}}{\varphi} \quad \text{for } \varphi < c_1 \tag{24}$$

with $k_1 = a_1\big(a_2 + \alpha(1 - a_2)\big)$, $c_1 = \frac{b_1\alpha}{1+b_2\alpha^{k_3}}$, $c_2 = \frac{4}{3}\alpha c_1^{-k_2} \cdot (c_3(1 - \alpha) + \alpha)$ where $a_1$ = 0.97, $a_2$ = 0.8, $b_1$ = 2.2455, $b_2$ = 3.45, $k_2$ = 1.25, $k_3$ = 1.29, $c_3$ = 2, and $\alpha$ is the aspect ratio of the ellipsoidal inclusions. At the disaggregation threshold found as $\varphi = c_1$ the partially molten material loses its
cohesiveness and both viscosities approach zero.

For a melt network consisting of 50% tubes and 50% films the following approximate equations have been derived from the model of Schmeling et al. (2012)

$$\eta_s = \eta_{s0} \cdot \left(1 - \frac{\varphi}{\varphi_{max}}\right)^{k} \tag{25}$$

$$\eta_b = \eta_{s0}a_2\left(\frac{\varphi_{max}-\varphi}{\varphi}\right)^{b_2} \tag{26}$$

The parameters needed to calculate these viscosities for different aspect ratios between 0.2 and 0.5 are given in Tab. 1. $k$ is given by $k = a_1\varphi + b_1$.





**Tab. 1: Parameters to calculate the viscosities for a melt network consisting of 50% tubes and 50% films using (25) and (26)**

| $\alpha$ | a1 | a2 | b1 | b2 | $\varphi_{max}$ |
|---|---|---|---|---|---|
| **0.2** | 0.8074 | 2.595 | 0.7009 | 1.276 | 0.2428 |
| **0.3** | 0.7435 | 2.622 | 0.7082 | 1.278 | 0.2629 |
| **0.4** | 0.6958 | 2.645 | 0.7145 | 1.281 | 0.2730 |
| **0.5** | 0.6692 | 2.664 | 0.7182 | 1.284 | 0.2785 |

Figure 1 shows the effective shear and bulk viscosities for different aspect ratios together with the simplified previous laws (1) and (2).

Takei and Holtzman (2009) and Rudge (2018) suggest that in the presence of an infinitesimal amount of connected melt the effective viscosity undergoes a finite drop of the order of a few 10% of the intrinsic matrix viscosity. In our approach we always have a finite melt porosity, thus we may identify the zero porosity viscosity $\eta_{s0}$ in our formulation with the initially weakened value of Takei and Holtzman (2009) or Rudge (2018).


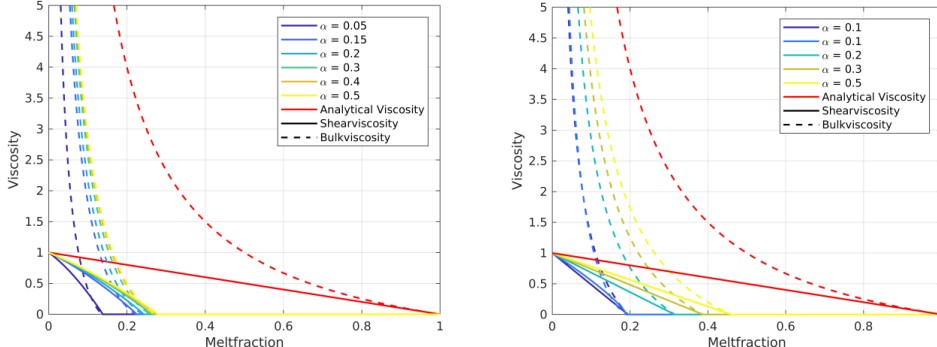

**Fig. 1.  Shear (solid) and bulk (dashed) viscosity for several aspect ratios as a function of the melt fraction. Left: The viscosities are calculated for a melt network consisting of 50% tubes and 50% films. Right: The network consists of 100% films. The red lines show the simplified analytical viscosities (Equ.**
**(1) and (2)).**



### 2.3 Methods and model setup

The model consists of a square box with a constant background porosity $\varphi_0$ superimposed initially by a bell shaped porosity wave at the position $x_0, z_0$. As the shape and width of a solitary wave with a certain

rheology law and amplitude is not known we use a Gaussian wave of the form

$$\varphi = A \cdot \exp\left(\left(\frac{x - x_0}{w}\right)^2 - \left(\frac{z - z_0}{w}\right)^2\right)$$

and vary the initial width $w$ of the wave.

At the sides of the box symmetric boundaries and at the top and the bottom free slip boundaries are used. The in- and outflow velocities of matrix and melt at the top and bottom are prescribed in terms of the analytical solution of the background porosity.

The equations are solved on a 201x201 grid by finite differences using the code FDCON (e.g. Schmeling et al., 2019). Resolution tests have been made with grids varying from 101x101 to 401x401. They show that after a short transient time the wave velocity and amplitude of the evolved porosity wave approach constant values in the limit of infinite resolution for all viscosity laws used. The subsequently observed slow variations of the wave velocity and amplitude along a quasi-steady state dispersion curve can be

attributed to numerical diffusion at finite grid resolution. The resolution test shows that 1) the quasi-steady state wave velocity and amplitude are of error order 1, and 2) the dispersion curves obtained on a 201x201 grid overestimate the extrapolated phase velocity values by about 10%. Time step resolution tests show that the long term temporal behavior of the porosity waves is significantly improved if the time steps are chosen smaller than approximately 0.2 times the Courant criterion.

The amplitude and wave velocity of the evolving porosity wave is obtained at every time step by quadratic interpolation of the porosity values on the FD grid and determining the value and velocity of the position of the maximum of the quadratic function. The resulting wave velocity shows small oscillations in time, which are probably due to the interaction of the 1[st] order error in time when solving equation (3) and (4) and the 2[nd] order error of the interpolation. These oscillations are smoothed by

applying a moving average including 50 neighboring points. The resulting time series of porosity amplitude and wave velocity can be plotted as a curve with time as curve parameter in an amplitude –





wave velocity plot. This curve can be understood as a dispersion curve because the phase velocity depends on amplitude and thus implicitly on the width or wavelength of the porosity wave.

For the model series presented below the width and the amplitude of the initial wave, the background
porosity and the rheology law have been varied. All models were carried out using n=2 and n=3 in the permeability-porosity law.

**3. Results**

**3.1 Dispersion curves for varied widths and amplitudes**

As the shape of a two-dimensional porosity wave for a certain wave amplitude is not known, the initial
width is varied. In Fig. 2a we show a porosity wave of amplitude 8 initially positioned at x = 0.5 and z = 0.2 (left) as it rises through the model box. In Fig. 2b a horizontal cross section through the maximum of an initial wave and the resulting solitary wave at a late stage is shown. During the early stage the wave gains some amplitude as the volume of an equivalent solitary wave with the same amplitude would be smaller for this example. Then the amplitude of the ascending wave slowly decreases again due to
numerical diffusion and the evolving phase velocity – amplitude curve describes the quasi-steady state dispersion relation. At this point the wave is expected to be a solitary wave. The shape of this wave resembles a Gaussian bell curve quite well but does not fit exactly. The upper part of the wave in this example fits very well while the lower part is slightly wider.

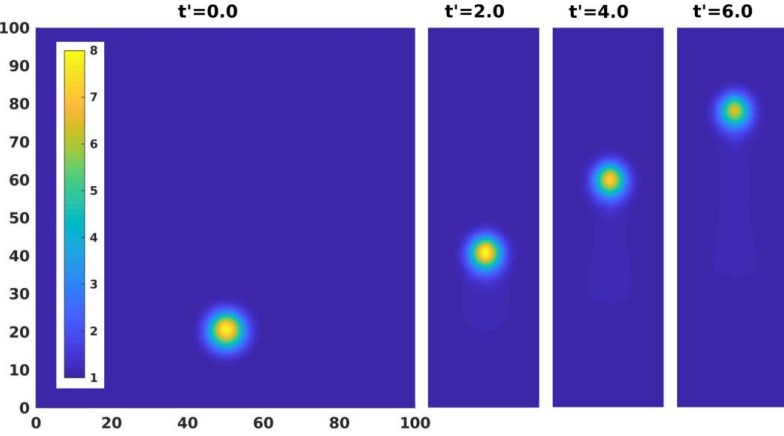





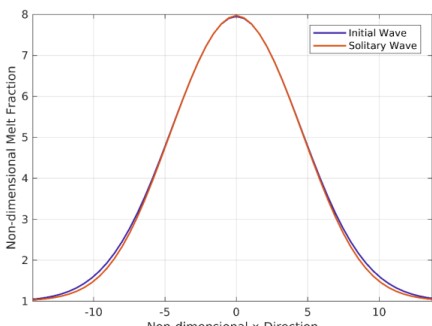

**Fig. 2. a) Non-dimensional melt fraction at 4 different time steps during the ascent of a solitary wave with an initial amplitude of 8. The model was carried out for a melt network geometry consisting of 100% films and an aspect ratio of 0.1. The background porosity is 0.005 and n = 3. b) Horizontal cross section through the center of the initial wave and the solitary wave at a later time.**

To analyze the evolution of the ascending solitary wave the phase velocity and the amplitude is tracked over the full rising time and plotted into a dispersion diagram. In Fig. 3 the dispersion curves of a model with a larger initial width than the resulting solitary wave, a model with a similar width, and a model with a smaller initial width are shown. The curves start with high velocities for the Gaussian bell shaped wave and then rapidly slow down until they approach a specific point visible as a sharp kink from which they slowly follow a line. For the bigger and optimal width models, after this kink the wave is expected to have reached the solitary wave stage. For the bigger initial width this stage is reached at a higher amplitude than initially assumed. It is important to note that, independent of the initial wave width, after reaching a solitary wave stage the velocities and shapes of waves of a certain amplitude are always equal, i.e. the three curves merge on one dispersion curve. For comparison with semi-analytic solitary porosity wave solutions the dashed curves in Fig. 3 and later Figures show dispersion curves with different power law n of the permeability-porosity relation and different bulk viscosity laws with m=0 assuming a constant bulk viscosity, and m=1 for a $1/\varphi$ proportionality (Simpson and Spiegelman, 2011). In contrast to our models these solutions a) use a stiff rheology ("analytic viscosity" in Fig 1), b) neglect solid shear (first term of the right hand side of equ. 8) which is responsible for $\vec{v}_1$ (c.f. equ. 12) in the matrix momentum equation, and for an important contribution in the separation flow (equ. 11), and c) apply the small porosity limit.



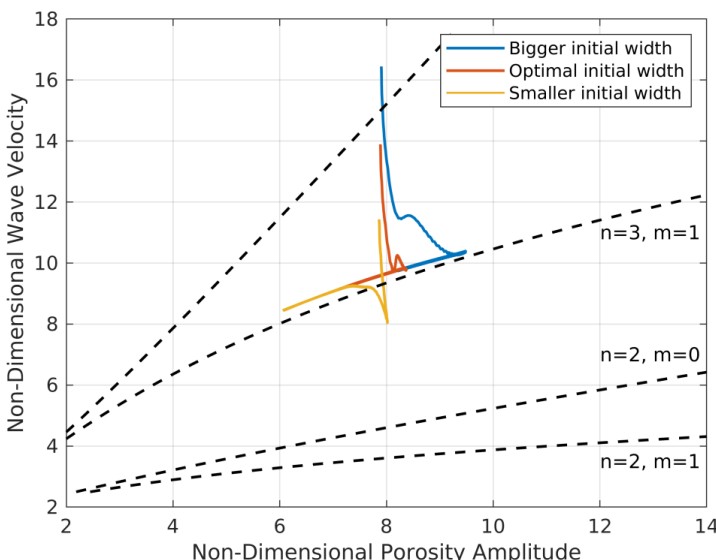

**Fig. 3. Dispersion curves for three models with an initial width bigger, smaller and approximately equal to the resulting solitary wave. Each model was carried out for a melt network geometry consisting of**

**100% films and an aspect ratio of 0.1. The background porosity is 0.005 and n = 3.**

Based on this result one can carry out many models with different initial wave widths and different initial amplitudes and get one empirical steady state solitary wave dispersion curve for one viscosity law for a wide range of amplitudes.

Fig. 4 shows the time-dependent dispersion curves of models with 4 different initial amplitudes (4 to 10),

and 11 different initial widths each. Depending on the initial widths they either gain amplitudes as they approach the solitary wave stage or they monotonously loose amplitude. Depending on the initial amplitude and width each case is characterized by a certain total melt volume, corresponding to a specific steady state solitary wave with a specific amplitude. Therefore the 44 models finally reach one steady state solitary wave dispersion curve at different amplitudes. As discussed in section 2, the

amplitude of the waves slowly continue to decrease due to some small amount of numerical diffusion. Yet, they continue following the steady state solitary wave dispersion curve.

Although we use a different rheology law and do not apply the simplifications mentioned above, the steady state dispersion curve of our model is in general agreement with the n = 3, m=1 dispersion curve determined semi-analytically by Simpson and Spiegelman (2011) (Fig. 4, dashed curve). However, given





the 10% numerical overestimation of phase velocities of our models (c.f. section 2.2), for high amplitudes
our dispersion curve shows a significantly smaller slope and correspondingly smaller phase velocities
than the semi-analytical curve by Simpson and Spiegelman (2011).

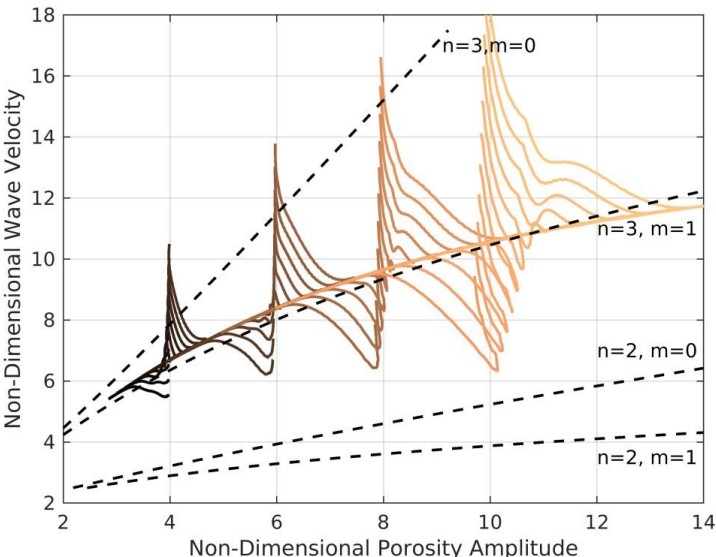

**Fig. 4. Dispersion curves for 44 models with 4 different initial amplitudes (4 to 10) and 11 different**
**initial widths each. All models were carried out for a melt network geometry consisting of 100% films**
**and an aspect ratio of 0.1. The background porosity is 0.005 and n = 3.**

### 3.2 Effect of different viscosity laws for n=2 and 3 on dispersion curves

To investigate the effect of different viscosity laws, two melt network geometries are chosen. The first
one consists of 50% films/ellipsoidal melt pockets and 50% tubes, the second of 100% films/ellipsoidal
melt pockets. Furthermore the aspect ratio α is varied, whereby a higher aspect ratio corresponds to
compact melt pockets and leads to stronger viscosities and to a higher disaggregation threshold (c.f. Fig.
1).

Waves with these different viscosity laws give only minor differences in the dispersion curves (Fig. 5a, b).
Especially with the films & tubes case the curves for different aspect ratios (Fig. 5a) are not
distinguishable, both during the transient and final stage. In contrast, the analytic viscosity case (equ. 1
and 2) propagates along a different path and converges to a 4 − 6 % higher final wave velocity curve.
With 100% Films the differences among curves with the different viscosity laws in the final velocity are



higher and lie in the order of 6 %. These differences are surprisingly small if compared to the actual differences in effective shear viscosities of about 13% and bulk viscosity of about a factor 4 (at 4 % melt
corresponding to a porosity amplitude 8). It is also to be noted that the steady state part of our dispersion curve calculated with the analytical viscosity (eq. 1 and 2) excellently agrees with the semi-analytical solution (dashed) by Simpson and Spiegelman (2011) for the same viscosity law, if we account for the 10 % numerical overestimation of our model phase velocity (c.f. section 2.2). Thus, their neglect of shear stresses and other simplifications have only a very minor effect compared to the effect of
different viscosity laws. The overall effect of weakening of matrix viscosity due to decreasing aspect ratio is to slow down the phase velocity slightly.

Changing n of the permeability-porosity relation to 2 decreases the wave velocities significantly (Fig.5c, d). This drop is consistent with the simplified semi-analytical solitary wave solutions (n=2, m=1, dashed curves). In contrast to the n=3 cases, the n=2 velocities are above the Simpson and Spiegelman (2011)
solutions even if the numerical 10 % overestimation is considered. As for the n=3 case, porosity waves with the stronger analytical viscosity case (equ. 1 and 2) are slightly faster than the new weaker viscosity cases.

While the ascending velocity of the wave is only slightly affected by the different viscositiy laws, the width of the wave changes more strongly. In Figure 6 the half-widths of the solitary waves of amplitude of 8 are plotted against the corresponding wave velocities for the different viscosity laws. For n=2 (Fig.
6a) and 100% films the wave gets wider for higher aspect ratios, while for the mixed geometry the widths stay more or less constant. The velocity increases only slightly with the aspect ratio. For n=3 (Fig. 6b) and 100% films the width increases with aspect ratio but in contrast to n=2 the phase velocity decreases with increasing aspect ratio. For the mixed geometry the velocity and half-width variations are
minor again. These results show that as long as melt tubes represent a significant portion of the total melt volume (here 50%) they control the porosity wave dynamics and keep the porosity wave properties rather fixed. Only in the absence of tubes compact melt pockets with large aspect ratios significantly broaden the waves. For the stiff case of analytical viscosity (equ. 1 and 2) the half width of the wave is comparable to the weaker 0.2 films, but the velocities are larger (Fig. 6a,b, light brown symbols).

Another interesting phenomenon to observe is the matrix velocity in the center of the wave, which increases for all geometries with aspect ratio (Fig. 7). While for 100% films this increase is stronger, for both geometries the velocities are approximately equal at an aspect ratio between 0.2 and 0.3. For n = 2 (Fig. 7a) the matrix velocities are always positive, meaning that despite a slow negative background



velocity of the matrix, it rises in the center of the wave (together with the melt). Interestingly, for n=3

(Fig. 7b) and small aspect ratios (0.1 and 0.2, i.e. weaker effective matrix viscosities) the direction of flow

of the matrix is changed and matrix in the center flows downwards, i.e. against the direction of melt

flow. Assuming constant matrix shear and bulk viscosities, Scott (1988) observed a similar switch from

negative to positive matrix velocities in the center of a 2D solitary wave when the ratio of the bulk to

shear viscosity was increased from 1 to 9 for n = 3. We see this switch around $\alpha$ = 0.25 corresponding to

a bulk to shear viscosity at the center of the porosity wave of about 16, and higher elsewhere. Such a

switch can be explained by an increasing role of diapiric flow, which is $\vec{v}_1$-related, incompressible, and

upward in the center of the wave, with respect to the compaction flow, which is $\vec{v}_2$-related, irrotational,

and downward in the center of the wave (c.f. equ. 12). Weakening of the bulk viscosity within the

porosity wave relative to the shear viscosity allows stronger decompaction and compaction rates which

amplify the downward compaction flow with respect to the upward diapiric flow.

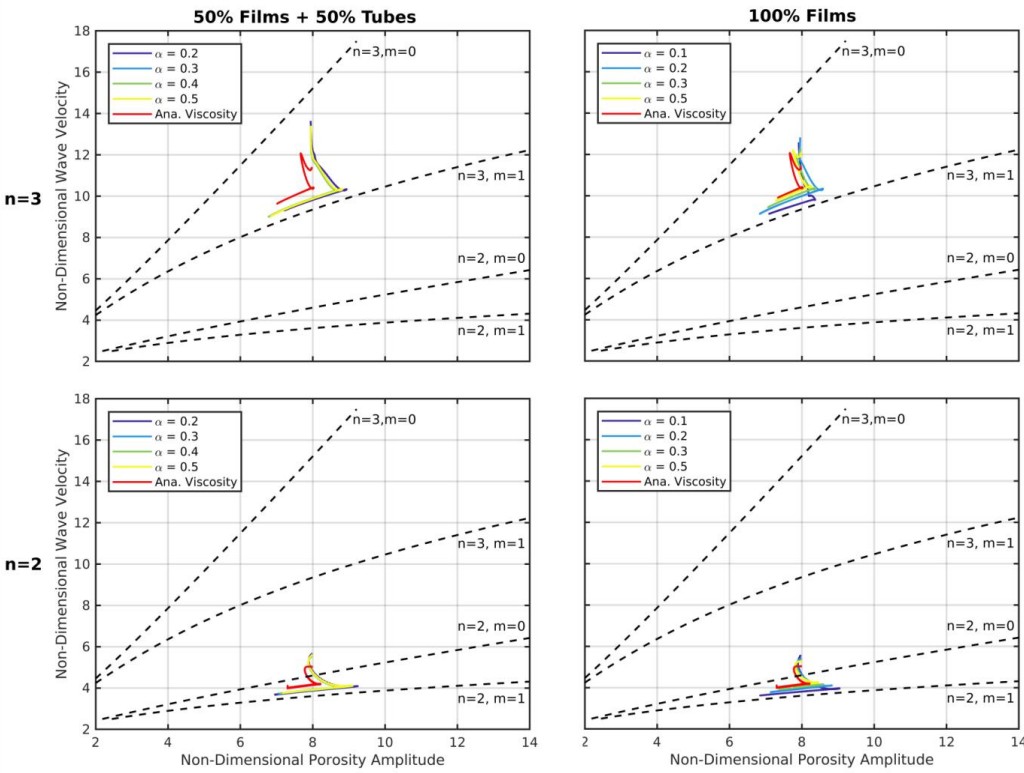

**Fig. 5. Dispersion curves of solitary waves with a) n=3, films & tubes, b) n=3, films, c) n=2, films & tubes, d) n=2, films for different aspect ratios.**





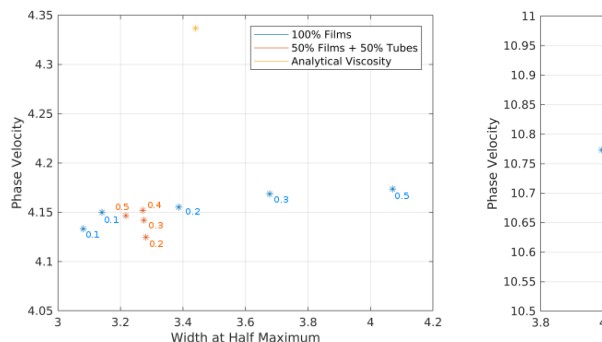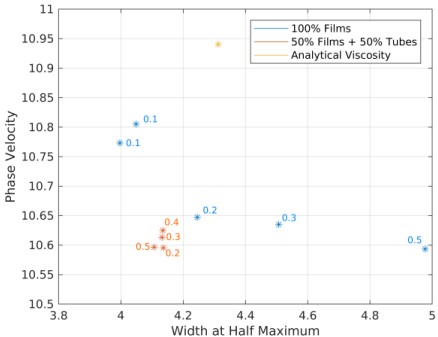

**Fig. 6. Non-dimensional half-width, plotted against non-dimensional wave velocity for a porosity wave of amplitude 8 for different viscosity laws. The numbers give the aspect ratios of the films/melt pockets. The background porosity is 0.5 %. a) Permeability-porosity exponent n = 2, b) n = 3**

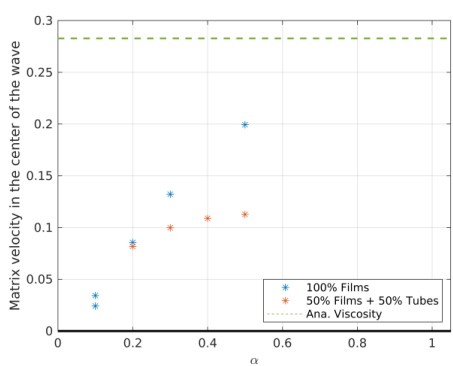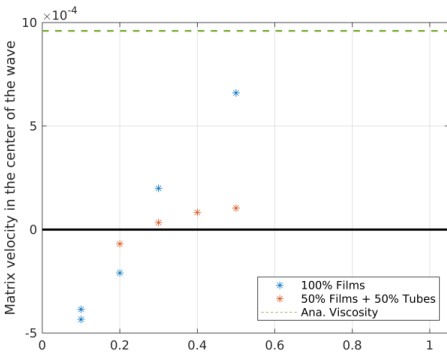

**Fig. 7. Matrix velocity in the center of a wave with an amplitude of 8 as a function of the aspect ratio**
**of the films for a) n = 2, b) n = 3. The background porosity for all models was 0.005.**

In the previous models the scaling background porosity of 0.005 and maximum wave amplitudes of 10 to 12 imply maximum melt fractions of 5 to 6 %. Thus the matrix shear viscosity decrease was only small, of order 10% for e.g. the aspect ratio 0.1 models and of order 5% for the stiffer analytical viscosity laws (1), (2). This explains the rather mild rheology effect when comparing the effect of the different viscosity

laws. With the aim to reach higher maximum melt fractions associated with stronger rheological effects we carried out a model series with increased background porosities, both applying the analytical viscosity law (m=1) and our weaker matrix viscosities with 100% films with an aspect ratio 0.1 (Fig. 8). The increase in the background porosity from 0.5 % to 1.5% has only a minor influence on the behavior of the solitary wave for models which use the analytical viscosity law (m=1): The half width of the wave is



almost completely unaffected (by ~ 1%), while the phase velocity is increased by only approximately

2.5%. Using a viscosity law based on a melt geometry consisting of 100% films and an aspect ratio of 0.1

the differences become more significant. The half width decreases to ~70% of its initial value and the

phase velocity decreases by up to 20% with increasing background porosity, i.e. with an increased

maximum porosity within the wave. Thus, the phase velocities show the opposite behavior to the

analytical viscosity law (see Fig. 8). These models suggest that the high melt fractions within the waves

which are associated with a significant local matrix weakening, both for shear and bulk viscosity, lead to

effectively shortened compaction lengths within the wave, i.e. to a narrowing and focusing of the wave.

Such narrower waves contain less melt than broader waves of same amplitude, i.e. less buoyancy, which

slows down the rising phase velocity.

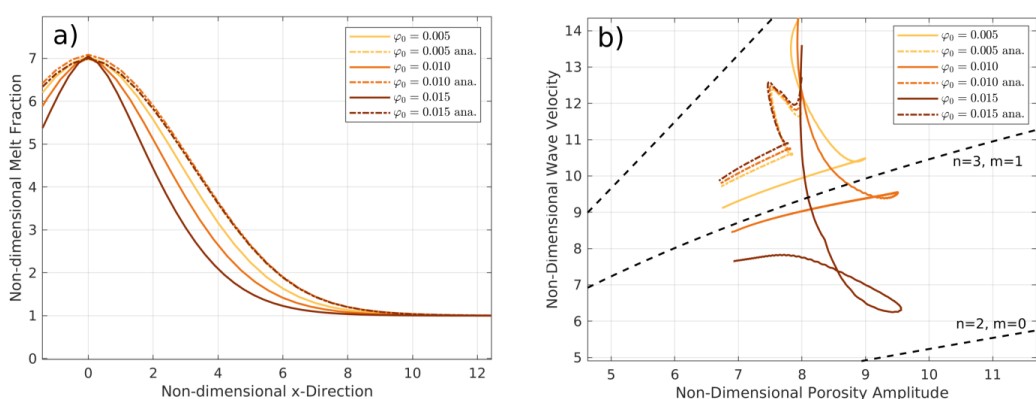


**Fig. 8. a) Horizontal profiles through ascending waves and b) dispersion curves with different background porosities but the same non-dimensional amplitude of 7. The dot-dashed curves were calculated with the simplified analytical viscosity law (m=1). The solid lines were calculated with a viscosity law based on 100% films and an aspect ratio of 0.1.**


### 4. Discussion

It is interesting to note that although the semi-analytic solutions of Simpson and Spiegelman (2011)

neglect the shear term in the matrix momentum equation and in the separation flow equation they are

in good agreement with the low $\varphi_0$ models which include this term. To understand this we made a test

with a model with 100% films and aspect ratio 0.1 and found that in the separation flow equation (11)

the shear term has a significant amplitude of about 50% compared to the compaction term. We then





switched off this term in the separation flow equation (11), which is equivalent to assuming zero shear viscosity. Surprisingly it turned out that separation velocity changed only insignificantly while the amplitudes of matrix divergence and convergence increases by about 25%, and the compaction related

term driving the separation velocity in equation (11) increases by about 50%, i.e. by the same amount the shear term had before. Obviously the buoyancy forces of the solitary wave are partitioned between the decompaction pressure controlled by the bulk viscosity and the shear stresses, namely the vertical normal shear stresses. If these stresses are neglected by assuming a zero shear viscosity, the buoyancy forces are balanced by the compaction pressure alone, and the shear contribution of the downward

segregation flow is taken over by the increased compaction contribution.

Recently, Rudge (2018) developed a diffusion creep model based on microscopic diffusion calculations in the presence of melt in textural equilibrium with truncated octahedrons. Assuming infinite diffusivity in the melt phase he obtains a somewhat stronger weakening of the shear viscosity at small melt fractions than in our model, but comparable disaggregation porosities as in Fig 1. However, due to the infinite

diffusivity assumption, the bulk viscosity remains finite (=5/3 of the effective shear viscosity) even at very small melt porosities, while in our model it increases infinitely in the limit of zero porosity. We expect that our results with increased weakening effect ($\varphi_0$ increased to 1.5%) might be applicable also to the rheology based on Rudge's (2018) analyses.

It should be noted that in our study the viscosity law has been varied by assuming various melt

geometries of melt films and films/melt pockets superimposed with tubes, while the permeability-porosity has been varied independently between $n = 2$ corresponding to the ideal case of only interconnected tubes and $n = 3$ corresponding to the ideal case of interconnected thin films. Three-dimensional melt distributions of partially molten mantle rocks have been studied e.g. by serial sectioning (Garapić et al. 2013) identifying a network of melt tubes and films, and by microtomography

(Zhu et al., 2011) suggesting the predominance of melt tubes along grain edges. Yet, at higher melt fractions the latter distributions are characterized by tapered edges of the melt tubes partly or completely wetting grain faces between adjacent grains. From the latter experiments Miller et al. (2014) determined the permeability by 3D-fluid flow modelling and found an exponent of 2.6. Thus, our simplified melt viscosities and permeabilities cover quite well observed partially molten olivine-basalt

systems in textural equilibrium.

In Richard et al. (2012) it was observed that with increasing background porosities the waves will widen and the phase velocities will slow down. In our models we observe faster velocities with increasing



background porosity if the analytical viscosity is used. This can be explained by the different scaling which was used by Richard et al. (2012). If the same scaling is used, we get the same behavior. In

contrast to Richard et al. (2012) we observe a narrowing effect of the waves for larger background porosities, which cannot be explained by scaling. As Richard et al. (2012) used a 1-D model, we suspect that 2-D effects such as including the incompressible flow velocity, $\vec{v}_1$, are responsible for the different shapes of the wave at different background porosities.

**5. Conclusion**

As the shape of a solitary wave in our models cannot be described analytically, we start with a Gaussian wave, which develops quite rapidly into a solitary wave with a similar shape and a certain amplitude, depending on the initial width of the wave.

Even though the rheologies used are much weaker than the simplified analytical ones the effect on dispersion curves and wave shape are only moderate as long as the maximum melt fractions do not

exceed about 20% of the disaggregation values. In this case the phase velocity changes just slightly for all cases, while the waves broaden in the absence of tubes with increasing aspect ratio. If the background porosities are increased such that the maximum melt fractions within the solitary waves reach values larger than about 50% of the disaggregation porosities, our models predict significant narrowing of the porosity waves and slowing down of the phase velocities. For such conditions a strong discrepancy in

solitary wave behavior between our viscosity law and the analytical ones is found.

For low melt fractions our models are in good agreement with semi-analytic solutions which neglect the shear stress term, because the matrix shear contribution of the downward segregation flow is taken over by the increase of the compaction contribution.

**Acknowledgements**

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
