# Peer review of "The effect of effective rock viscosity on 2D magmatic porosity waves"

_Solid Earth, 2019_

## Referee Comment (RC1) · Guillaume Richard (Referee) · 5 Aug 2019

The authors of the manuscript have used numerical modeling to tackles the question of the rheology of a two-phase magmatic mush. The question they want to answer is fully in the scope of Solid Earth. They built on their previous study on the question (Schmeling et al, 2012) and go further by testing the effects of the viscosity laws they have proposed on the propagation of solitary porosity waves in a 2D setting. They can conclude that even considering "realistic" rheologies, the transport of melt through porosity waves is a likely process in magmatic mushes. They well describe their methods and assumptions and in this framework their results seems sounds. I believe nevertheless that the manuscript could be improved by clarifying some parts and adding more developments (see my listing below).

Line 121. The chosen boundary conditions are neglecting the effect of the propagating wave. I agree that as long as the wave is far from the boundary this effect can be neglected but the question is always "how far is far enough ?". A more developed paragraph on this point would improve the trust of the readers on the model results. Maybe the comment "The upper part of the wave in this example fits very well while the lower part is slightly wider." (line 228) is related to boundary affects.

Sometimes the author has used the term "phase velocity" and sometimes he has used "wave velocity/velocity of the wave" . This makes the discussion more difficult to follow and should be clarified.

Line 405. It is stated that the effect of the background porosity is different from results of 1D model (Richard et al, 2012): high background porosity inducing narrower waves instead of larger ones. I don't understand this difference and I think that this manuscript would be the right place to investigate this "dimensional" effect.

Lines 415 & 418. Results are discussed in terms of matrix disaggregation threshold. As seen in Schmeling et al. (2012) the disaggregation porosity is strongly model dependent and thus should be used carefully. In the conclusion, I would suggest to give more details on the input parameters and on the actual value of the disaggregation porosity of the cases that are discussed.

---

## Referee Comment (RC2) · Viktoriya Yarushina (Referee) · 27 Sep 2019

Bulk and shear viscosities of the porous rocks are dynamic quantities that change in response to pressure, temperature and porosity variations. Various laws for porosity dependence of the bulk viscosity on porosity were proposed in the literature. Some of them differ significantly from each other and sometimes even show opposite trends [Yarushina and Podladchikov, 2015] (Fig. 1). On the other hand, previous research showed that porosity waves change their properties significantly depending on the rock rheology and dependence of bulk viscosity on effective pressure [e.g., Connolly and Podladchikov, 2007; Yarushina et al., 2015]. Thus, it is very important to explore how various expressions for porosity dependence of effective viscosity influence propagation of porosity waves. Manuscript of Dohmen et al. addresses this issue. The authors

investigate how porosity waves change their properties based on the effective viscosity relation previously proposed by the authors [Schmeling et al., 2012]. They compare results of 2D simulations with previously published 1D results for different types of the bulk viscosity with a conclusion that exact choice of porosity dependence of bulk and shear viscosities play a minor role on the speed of porosity wave propagation for the cases considered. The manuscript presents new concepts with interesting conclusions and, thus, deserves the publication in the journal.

Some points for improvement:

- References in the manuscript do not sufficiently cover recent work and sometimes misleading. For example, lines 46-51 state "Within supersolidus source regions at low melt fractions melt is assumed to slowly percolate by two-phase porous flow within a deforming matrix (McKenzie, 1984; Schmeling, 2000; Bercovici et al., 2001), followed by melt accumulation within rising high porosity waves (Scott and Stevenson, 1984; Spiegelman, 1993, Wiggins and Spiegelman, 1995; Richard et al., 2012) or focusing into channels (Stevenson, 1989; Richardson, 1998) which have the potential to pene-trate into subsolidus regions above to generate dykes."

However, the work of Scott and Stevenson [1984] does not present focusing of the melt into channels. It discusses linear stability analysis, which gives conditions at which flow instability may arise. Subsequently, it was shown by various authors that this instability may result in different 3D shapes including spherical blobs and sills [e.g. Wiggins and Spiegelman, 1995]. On the other hand, formation of 3D channels due to two-phase porous flow within a deforming matrix were demonstrated in [Omlin et al., 2018; Räss et al., 2014].

- Similarly, lines 54-57 state that "So far most of the porosity wave model approaches used either equal bulk and shear viscosities, or simple laws in the form of [eqn (1) and (2)]", where equations (1) and (2) represent simple porosity dependent quantities. This is an incorrect statement – most model approaches but not most recent ones.

In fact, much more complicated relations for pressure-dependent weakening viscosity were considered in some recent works of [Omlin et al., 2018; Yarushina et al., 2015]. Moreover, Omlin et al. [2018] study the effect of the ratio of bulk and shear viscosities on the speed of wave propagation. The implication of this to magmatic systems can be important.

- Equations (3) and (4). Do you really need to keep  $\rho_f$  and  $\rho_s$  here?

- Lines 87 – 93. It is important to emphasize already here that P is the FLUID pressure and  $\tau_{ij}$  is the EFFECTIVE stress tensor to avoid possible confusion. In 2-phase system there are many different pressures and stress tensors, i.e. solid stress, total stress, effective stress.

- Lines 94 – 101. As authors state earlier in line 84, solid and fluid densities are constant. Thus, it is a bit odd to read about linearized equation of state for fluid and mixture densities in lines 96-97. Mixture density is already linear with respect to porosity:  $\rho = \rho_f \phi + \rho_s (1 - \phi)$ . The real meaning of equations (9) and (10) is introduction of new unknowns  $\rho_0$ , which is nothing else than  $\rho_s$  and  $c_f$ . Introduction of these new parameters is unnecessary and complicates reading a paper, which is already mathematically heavy.

- Lines 102 - 103. Statement "Neglecting capillary pressure at the melt-solid interfaces, the pressure is equal in the melt phase and in the solid phase" is wrong. Difference between solid and fluid pressures does not stem from capillary forces. What authors might want to say here is that parameter P in equations (5) and (6) is the same and is the fluid pressure. This confusion may be avoided if authors will label variables as suggested before.

- Equations (16), (17). Do you mean  $k_0$  instead of  $k_{\phi}$  here?

- Lines 188-189. Unclear sentence. Please, rephrase.

- Line 196 – 198. "They show that after a short transient time the wave velocity and

amplitude of the evolved porosity wave approach constant values in the limit of infinite resolution for all viscosity laws used." You do not have infinite resolution, please rephrase.

- Lines 236-237. "In Fig. 3 the dispersion curves of a model with a larger initial width than the resulting solitary wave..." Initial width of what? Please, be clear.

- Lines 246-248. Please, explain new variable m. Reading of the paper will be significantly easier if you would provide expressions for the bulk viscosity that was used for comparison with your simulations.

- Section 3.2. For each values of parameters that you discuss, it would be nice to see how far different bulk viscosities lie from each other.

- Lines 267 – 275. Authors compare their numerical solution with published analytical solution obtained in a small porosity limit and conclude that there is a deviation from analytical solution at larger values of amplitude. There could be various reasons for this deviation, including numerical error and limitations of the analytical solution as porosity grows higher. To rule out the second reason for discrepancy, it would be useful to compare results of numerical simulations, which are free of the assumption of small porosity, with full analytical solutions such as in [eqn (38), Yarushina et al., 2015].

- Authors conclude that agreement with analytical solution is reasonable. However, comparison was made only for moderate porosity waves with amplitudes below 14, where porosity within the channel reached only 7%. Waves with higher amplitude, which bring more significant amount of melt, will deviate much stronger. The melt fraction in the mantle maybe mostly below 7%. However, for results to have a more general application, the investigated porosity range could have been extended.

- Lines 403-406. "This can be explained by the different scaling which was used by Richard et al. (2012). If the same scaling is used, we get the same behavior. In contrast to Richard et al. (2012) we observe a narrowing effect of the waves for larger

background porosities, which cannot be explained by scaling. As Richard et al. (2012) used a 1-D model". Richard et al. used different expression for compaction length that contained porosity and thus, comparison, of course should take this into account. Please, be a bit more specific on what you mean by "different scaling". It would be useful to show some illustration to what kind of differences you see between your solution and previous result of Richard et al.

- Line 415. "exceed about 20% of the disaggregation values." It is better to use exact values here. Your disaggregation values are internal model parameter.

**References**

Connolly, J. A. D., and Y. Y. Podladchikov (2007), Decompaction weakening and channeling instability in ductile porous media: Implications for asthenospheric melt segregation, J Geophys Res-Sol Ea, 112(B10), B10205, doi:10.1029/2005jb004213.

Omlin, S., L. Rass, and Y. Y. Podladchikov (2018), Simulation of three-dimensional viscoelastic deformation coupled to porous fluid flow, Tectonophysics, 746, 695-701, doi:10.1016/j.tecto.2017.08.012.

Räss, L., V. M. Yarushina, N. S. C. Simon, and Y. Y. Podladchikov (2014), Chimneys, channels, pathway flow or water conducting features - an explanation from numerical modelling and implications for CO2 storage, Energy Procedia, 63, 3761-3774, doi:10.1016/j.egypro.2014.11.405.

Schmeling, H., J. P. Kruse, and G. Richard (2012), Effective shear and bulk viscosity of partially molten rock based on elastic moduli theory of a fluid filled poroelastic medium, Geophys J Int, 190(3), 1571-1578, doi:10.1111/j.1365-246X.2012.05596.x.

Scott, D. R., and D. J. Stevenson (1984), Magma solitons, Geophysical Research Letters, 11(11), 1161-1164.

Yarushina, V. M., and Y. Y. Podladchikov (2015), (De)compaction of porous viscoelastoplastic media: Model formulation, Journal of Geophysical Research Solid Earth, 120,

**C5**

doi:10.1002/2014JB011258.

Yarushina, V. M., Y. Y. Podladchikov, and J. A. D. Connolly (2015), (De)compaction of porous viscoelastoplastic media: Solitary porosity waves, Journal of Geophysical Research Solid Earth, 120, doi:10.1002/2014JB011260.

Fig. 1. Figure 1. Compilation plot of selected effective viscosities presented in the literature (from Yarushina and Podladchikov [2015])

---

## Author Comment (AC1) · 24 Oct 2019

We would like to thank both reviewers for the very thoughtful and detailed comments, which have helped to improve our manuscript. Please find below a point-to-point reply to both comments. The review comments are in black, our answers in red and the changes in the manuscript in green.

**Review 1: Guillaume Richard**

The authors of the manuscript have used numerical modeling to tackles the question of the rheology of a two-phase magmatic mush. The question they want to answer is fully in the scope of Solid Earth. They built on their previous study on the question (Schmeling et al, 2012) and go further by testing the effects of the viscosity laws they have proposed on the propagation of solitary porosity waves in a 2D setting. They can conclude that even considering "realistic" rheologies, the transport of melt through porosity waves is a likely process in magmatic mushes. They well describe their methods and assumptions and in this framework their results seems sounds. I believe nevertheless that the manuscript could be improved by clarifying some parts and adding more developments (see my listing below).
121. The chosen boundary conditions are neglecting the effect of the propagating wave. I agree that as long as the wave is far from the boundary this effect can be neglected but the question is always "how far is far enough ?". A more developed paragraph on this point would improve the trust of the readers on the model results.

Several sizes for the ascending wave in the model have been tested and it could be shown that if the wave is small enough the effect of the boundaries on the ascending velocity and size of the wave can be neglected. In the dispersion relation figure one can see that as the wave approaches the upper boundary the dispersion relation slightly deviates from the supposed line. This is the error due to the boundary and is smaller than 0.5% as long as the distance to the boundary is smaller than 1.5 times the 10%-radius of the wave.
The mirror side boundaries have just a minor influence. As long as these boundaries are further away than 3 times the 10%-radius the error in velocity should be less than 1%. In our models the waves have a distance of 7-10 times the 10%-radius which leads to an error smaller than 0.2%.

New paragraph: "The influence of the boundaries on the ascending wave was investigated and found to be fairly small. In Fig. 3 one can see the effect of the upper boundary on the phase velocity. At the end, as the waves approach the upper boundary, the dispersion curves slightly deviate from the supposed line. This error is smaller than 0.5% as long as the distance of the center of the wave to the upper boundary is greater than 1.5 times its 10%-radius. This radius is defined as the radius at which the porosity has decreased to 10% of the amplitude of the wave. For the side boundaries this distance has to be larger. For distances greater than 3 times the 10%-radius this error is smaller than 1%. In our models the waves have distances of 7-10 times the 10%-radius which corresponds to errors between 0.2 and 0.05%."

Maybe the comment "The upper part of the wave in this example fits very well while the lower part is slightly wider." (line 228) is related to boundary affects.

It is hard to find the exact effect of the boundaries on the shape of the wave but as the effect on the velocity is fairly small I would expect this effect to be in the same order of magnitude. Anyways, I would suspect that the boundaries would concentrate the wave as it limits the area.

Nothing has been changed.

Sometimes the author has used the term "phase velocity" and sometimes he has used "wave velocity/velocity of the wave" . This makes the discussion more difficult to follow and should be clarified.

This is correct and will be changed.

All "wave velocity/velocity of the wave" have been changed to "phase velocity".

Line 405. It is stated that the effect of the background porosity is different from result of 1D model (Richard et al, 2012): high background porosity inducing narrower waves instead of larger ones. I don't understand this difference and I think that this manuscript would be the right place to investigate this "dimensional" effect.

Yes, I understand your concern but we do not fully understand it either. We were kind of surprised when we observed this contrary behavior and to better understand this one would have to carry out 1-D models in our code to prove that it is really a "dimensional" effect. But as the scope of this paper is not the comparison of 1 to 2-D porosity waves and it would not be easy to set up 1-D models in our code in such a short time we further have to suspect that it is a dimensional effect. Anyways, to better understand what the effect of the different scalings is we will add Fig. 8 with the scaling of Richard et al. (2012) to the supplementary.

The difference in scaling is explained, Fig. 8 but with the scaling used in Richard et al. (2012) was added to the supplementary.

Lines 415 & 418. Results are discussed in terms of matrix disaggregation threshold. As seen in Schmeling et al. (2012) the disaggregation porosity is strongly model dependent and thus should be used carefully. In the conclusion, I would suggest to give more details on the input parameters and on the actual value of the disaggregation porosity of the cases that are discussed.

Yes, it would be better to give some more information on the viscosities of the cases discussed. This could also help to make it better understandable.

The corresponding paragraph has been partly rephrased:
"Even though the rheologies used are much weaker than the simplified analytical ones the effect on dispersion curves and wave shape are only moderate as long as the shear viscosity does not drop below about 80% of the intrinsic shear viscosity. This value corresponds to a melt fraction of 5 %, equivalent to 20% of the disaggregation value. At this porosity the bulk viscosity is approximately 5-7 times the intrinsic shear viscosity. In this case the phase velocity changes just slightly for all cases, while the waves broaden in the absence of tubes with increasing aspect ratio. If the shear viscosity has been decreased to 50% of the intrinsic viscosity, which corresponds to melt fractions of 12% equivalent to 50% of the disaggregation values and to small bulk viscosities in the order of the intrinsic shear viscosity, our models predict significant narrowing of the porosity waves and slowing down of the phase velocities. For such conditions a strong discrepancy in solitary wave behavior between our viscosity law and the analytical ones is found."

**Review 2: Viktorya Yarushina**

Bulk and shear viscosities of the porous rocks are dynamic quantities that change in response to pressure, temperature and porosity variations. Various laws for porosity dependence of the bulk viscosity on porosity were proposed in the literature. Some of them differ significantly from each other and sometimes even show opposite trends [Yarushina and Podladchikov, 2015] (Fig. 1). On the other hand, previous research showed that porosity waves change their properties significantly depending on the rock rheology and dependence of bulk viscosity on effective pressure [e.g., Connolly and Podladchikov, 2007; Yarushina et al., 2015]. Thus, it is very important to explore how various expressions for porosity dependence of effective viscosity influence propagation of porosity waves. Manuscript of Dohmen et al. addresses this issue. The authors investigate how porosity waves change their properties based on the effective viscosity relation previously proposed by the authors [Schmeling et al., 2012]. They compare results of 2D simulations with previously published 1D results for different types of the bulk viscosity with a conclusion that exact choice of porosity dependence of bulk and shear viscosities play a minor role on the speed of porosity wave propagation for the cases considered. The manuscript presents new concepts with interesting conclusions and, thus, deserves the publication in the journal.

Some points for improvement:

- References in the manuscript do not sufficiently cover recent work and sometimes misleading. For example, lines 46-51 state "Within supersolidus source regions at low melt fractions melt is assumed to slowly percolate by two-phase porous flow within a deforming matrix (McKenzie, 1984; Schmeling, 2000; Bercovici et al., 2001), followed by melt accumulation within rising high porosity waves (Scott and Stevenson, 1984; Spiegelman, 1993, Wiggins and Spiegelman, 1995; Richard et al., 2012) or focusing into channels (Stevenson, 1989; Richardson, 1998) which have the potential to penetrate into subsolidus regions above to generate dykes." However, the work of Scott and Stevenson [1984] does not present focusing of the melt into channels. It discusses linear stability analysis, which gives conditions at which flow instability may arise. Subsequently, it was shown by various authors that this instability may result in different 3D shapes including spherical blobs and sills [e.g. Wiggins and Spiegelman, 1995]. On the other hand, formation of 3D channels due to two-phase porous flow within a deforming matrix were demonstrated in [Omlin et al., 2018; Räss et al., 2014].

Yes, you are right. Thank you for the correction. The mentioned papers will be added and shortly discussed.

The corresponding paragraph has been changed:
"Within supersolidus source regions at low melt fractions melt is assumed to slowly percolate by two-phase porous flow within a deforming matrix (McKenzie, 1984; Schmeling, 2000; Bercovici et al., 2001), followed by melt accumulation within rising high porosity waves (Scott and Stevenson, 1984; Spiegelman, 1993, Wiggins and Spiegelman, 1995; Richard et al., 2012) or focusing into channels which can possibly penetrate into subsolidus regions. Thereby Stevenson (1989) did a linear stability analysis to find conditions at which flow instabilities may arise, which may result in different 3D shapes (Richardson, 1998; Wiggins and Spiegelman, 1995). Formation of 3D channels within a deforming matrix have been demonstrated in Omlin et al. (2018) or Räss et al. (2014)."

- Similarly, lines 54 – 57 state that "So far most of the porosity wave model approaches used either equal bulk and shear viscosities, or simple laws in the form of [eqn (1) and (2)]", where equations (1) and (2) represent simple porosity dependent quantities. This is an incorrect statement – most model approaches but not most recent ones. In fact, much more complicated relations for pressure-dependent weakening viscosity were considered in some recent works of [Omlin et al., 2018; Yarushina et al.,

2015]. Moreover, Omlin et al. [2018] study the effect of the ratio of bulk and shear viscosities on the speed of wave propagation. The implication of this to magmatic systems can be important.

The mentioned papers use a pressure dependent power law formulation for the viscosity but still use simplified laws for the porosity-dependence of the viscosities. But still, these papers should be mentioned and discussed.

"There are also recent models that use pressure dependent weakening viscosities but still use the simplified equations mentioned above for the porosity dependence of the viscosity (Omlin et al., 2018; Yarushina et al., 2015)." was added.

The Equations (3) and (4). Do you really need to keep $\rho$ f and $\rho$ s here?

Yes, you are right. We don't need to keep the densities here and are just a relic due to the neglect of the source terms on the right-hand-side we usually solve.

The densities in the equations (3) and (4) have been erased.

- Lines 87 – 93. It is important to emphasize already here that P is the FLUID pressure and $\tau$ ij is the EFFECTIVE stress tensor to avoid possible confusion. In 2-phase system there are many different pressures and stress tensors, i.e. solid stress, total stress, effective stress.

Yes, you are right. It would be better understandable to use these specifications.

"effective viscous stress tensor of the matrix" and "fluid" was added.

- Lines 94 – 101. As authors state earlier in line 84, solid and fluid densities are constant. Thus, it is a bit odd to read about linearized equation of state for fluid and mixture densities in lines 96-97. Mixture density is already linear with respect to porosity: $\rho = \rho$ f $\varphi + \rho$ s $(1 − \varphi)$. The real meaning of equations (9) and (10) is introduction of new unknowns $\rho$ 0 , which is nothing else than $\rho$ s and c f . Introduction of these new parameters is unnecessary and complicates reading a paper, which is already mathematically heavy.

Thank you for mentioning this. The rho_0 is just an artifact from the not isothermal models we usually solve. In this case, where the temperature has no influence on the density we do not need rho_0. We will change this part of the paper and adjust the equations.

The equations of state for the densities were replaced by $\rho = \rho_f\varphi + \rho_s(1 − \varphi)$. $\rho_0 c_f$ in now equation (10) has been replaced by $(\rho_s − \rho_f)$. $(\rho_0 − \rho_{0f})$ in now equation (16) has been replaced by $(\rho_s − \rho_f)$. $\rho_{s0}$ in now equation (17) has been changed to $\rho_s$.

- Lines 102 – 103. Statement "Neglecting capillary pressure at the melt-solid interfaces, the pressure is equal in the melt phase and in the solid phase" is wrong. Difference between solid and fluid pressures does not stem from capillary forces. What authors might want to say here is that parameter P in equations (5) and (6) is the same and is the fluid pressure. This confusion may be avoided if authors will label variables as suggested before.

Yes, you are right. This sentence is wrong and will be corrected.

The parameters have been labeled as shown before and the mentioned sentence has been changed to "The fluid pressure in equation (5) and (6) is the same and can be eliminated by merging the two equations."

- Equations (16), (17). Do you mean k 0 instead of k φ here?

No, we mean  k φ at the reference state.  For the compaction length we use the specific permeability according to equation (7). There are many different ways to scale porosity waves and it is really confusing if you want to compare results. Some scalings even lead to opposite behaviors. Concerning the porosity dependency of the compaction length, there are many who use k_0 (e.g. Connolly & Podladchikov, 2007; Yarushina et al., 2015; Sramek et al., 2007; Barcilon & Richter, 1986) but there are also many who use the porosity dependent one (Omlin et al., 2018; Richard et al., 2012; Cai & Bercovici, 2016; Rabinowicz et al., 2002; Simpson & Spiegelman, 2011; Scott & Stevenson, 1986). We use the porosity dependent one like it was calculated by McKenzie (1984).

We reformulate the scaling procedure in a more consistent way, emphasizing the reference state.

- Lines 188-189. Unclear sentence. Please, rephrase.

The sentence has been changed to "For the model we use a square box (1x1), which is initially partially molten to a certain degree, the background porosity. We place an initial porosity anomaly with a higher porosity centered at x_0=0.5 and z_0=0.2 from which a porosity wave will develop."

- Line 196 – 198. "They show that after a short transient time the wave velocity and amplitude of the evolved porosity wave approach constant values in the limit of infinite resolution for all viscosity laws used." You do not have infinite resolution, please rephrase.

The meaning of this sentence is that if we would decrease dx and dz further, the change in amplitude and velocity would converge nearer to 0 and would finally reach 0 in the case of infinite small dx and dz. The sentence will be rephrased.

"in the limit of infinite resolution" was changed to "for very high resolutions".

- Lines 236-237. "In Fig. 3 the dispersion curves of a model with a larger initial width than the resulting solitary wave. . ." Initial width of what? Please, be clear.

The initial width of the gaussian bell-shaped starting wave. The sentence will be rephrased.

The sentence was rephrased to
"In Fig. 3 the dispersion curves of a model with a starting wave width which is initially larger than the resulting solitary wave …"

- Lines 246-248. Please, explain new variable m. Reading of the paper will be significantly easier if you would provide expressions for the bulk viscosity that was used for comparison with your simulations.

Yes, you are right. m should have been already in equation (2) but it's missing and will be added.

m has been added to equation (2) and is now $\eta_b = \eta_{s0} C \frac{(1-\varphi)}{\varphi^m}$ and an explanation has been added: "and m=0 for equal shear and bulk viscosities or m=1 otherwise".

- Section 3.2. For each values of parameters that you discuss, it would be nice to see how far different bulk viscosities lie from each other.

We don't think that we learn much from plotting or mentioning the bulk viscosity for every model. Only for the models with different background porosity, i.e. large absolute porosities,  we see an interesting correlation between phase velocities and bulk viscosities. We now mention this:

"If the shear viscosity has been decreased to 50% of the intrinsic viscosity, which corresponds to melt fractions of 12% equivalent to 50% of the disaggregation values and to small bulk viscosities in the order of the intrinsic shear viscosity, our models predict significant narrowing of the porosity waves and slowing down of the phase velocities".

- Lines 267 – 275. Authors compare their numerical solution with published analytical solution obtained in a small porosity limit and conclude that there is a deviation from analytical solution at larger values of amplitude. There could be various reasons for this deviation, including numerical error and limitations of the analytical solution as porosity grows higher. To rule out the second reason for discrepancy, it would be useful to compare results of numerical simulations, which are free of the assumption of small porosity, with full analytical solutions such as in [eqn (38), Yarushina et al., 2015].

We compared the 1-D solution of Simpson and Spiegelman (2011) with the full solution of Yarushina et al. (2015) and could observe a decreasing slope for the full solution compared to the simplified one, wich is quite similar to what we observe in our 2-D models.

"Comparison of the simplified semi-analytical 1-D solution of Simpson and Spiegelman (2011) with the full analytical 1-D solution of Yarushina et al. (2015) shows that for low porosities these solutions fit very well together. For higher porosities the full solution becomes slower than the simplified one. Tentatively transferring this result to 2D our decrease in the slope can probably be explained by the low porosity limitation of the Simpson and Spiegelman (2011) solution which overestimates the velocity at high porosities" was added.

- Authors conclude that agreement with analytical solution is reasonable. However, comparison was made only for moderate porosity waves with amplitudes below 14, where porosity within the channel reached only 7%. Waves with higher amplitude, which bring more significant amount of melt, will deviate much stronger. The melt fraction in the mantle maybe mostly below 7%. However, for results to have a more general application, the investigated porosity range could have been extended.

Perhaps the reviewer has missed the section in which we discuss the models with higher porosity, which significantly deviate from the analytical viscosity law.  It was kind of hidden…We now emphasize this point by an additional statement, explicitly mention the higher porosities, and reorganize the conclusions.

New sentence: "Thus, the half widths and phase velocities show a significant difference to the analytical viscosity law (Fig. 8)."
Conclusion reorganized: "In contrast, for higher melt fractions of about 12%, equivalent to 50% of the disaggregation values, the shear viscosity decreases to 50% of the intrinsic viscosity, and the bulk

viscosities is of the order of the intrinsic shear viscosity. Then, our models predict significant narrowing of the porosity waves and slowing down of the phase velocities. For such conditions a strong discrepancy in solitary wave behavior between our viscosity law and the analytical ones is found."

- Lines 403-406. "This can be explained by the different scaling which was used by Richard et al. (2012). If the same scaling is used, we get the same behavior. In contrast to Richard et al. (2012) we observe a narrowing effect of the waves for larger background porosities, which cannot be explained by scaling. As Richard et al. (2012) used a 1-D model". Richard et al. used different expression for compaction length that contained porosity and thus, comparison, of course should take this into account. Please, be a bit more specific on what you mean by "different scaling". It would be useful to show some illustration to what kind of differences you see between your solution and previous result of Richard et al.

Yes, we clarify what we mean with "different scaling" but the difference is not the porosity dependence as we also use a porosity dependent compaction length but Richard et al. (2012) uses just the shear viscosity and not the bulk viscosity in his compaction length.

Additional sentence:"They used just the shear viscosity to calculate the compaction length and not the sum of shear and bulk viscosity. ". Fig. 8 but with the Richard et al. (2012) was added to the supplementary.

- Line 415. "exceed about 20% of the disaggregation values." It is better to use exact values here. Your disaggregation values are internal model parameter.

The disaggregation values are dependent on the used viscosity law and vary between 10 and 40%. 20% of these porosities would be therefore between 2 and 8%.

The corresponding paragraph has been rephrased:
"Even though the rheologies used are much weaker than the simplified analytical ones the effect on dispersion curves and wave shape are only moderate as long as the shear viscosity does not drop below about 80% of the intrinsic shear viscosity. This value corresponds to a melt fraction of 5 %, equivalent to 20% of the disaggregation value. At this porosity the bulk viscosity is approximately 5-7 times the intrinsic shear viscosity. In this case the phase velocity changes just slightly for all cases, while the waves broaden in the absence of tubes with increasing aspect ratio.
   In contrast, for higher melt fractions of about 12%, equivalent to 50% of the disaggregation values, the shear viscosity decreases to 50% of the intrinsic viscosity, and the bulk viscosities is of the order of the intrinsic shear viscosity. Then, our models predict significant narrowing of the porosity waves and slowing down of the phase velocities. For such conditions a strong discrepancy in solitary wave behavior between our viscosity law and the analytical ones is found. "

References
Connolly, J. A. D., and Y. Y. Podladchikov (2007), Decompaction weakening and channeling instability in ductile porous media: Implications for asthenospheric melt segregation, J Geophys Res-Sol Ea, 112(B10), B10205, doi:10.1029/2005jb004213.
Omlin, S., L. Rass, and Y. Y. Podladchikov (2018), Simulation of three-dimensional viscoelastic deformation coupled to porous fluid flow, Tectonophysics, 746, 695-701, doi:10.1016/j.tecto.2017.08.012.

Räss, L., V. M. Yarushina, N. S. C. Simon, and Y. Y. Podladchikov (2014), Chimneys, channels, pathway flow or water conducting features - an explanation from numerical modelling and implications for CO2 storage, Energy Procedia, 63, 3761-3774, doi:10.1016/j.egypro.2014.11.405.

Schmeling, H., J. P. Kruse, and G. Richard (2012), Effective shear and bulk viscosity of partially molten rock based on elastic moduli theory of a fluid filled poroelastic medium, Geophys J Int, 190(3), 1571-1578, doi:10.1111/j.1365-246X.2012.05596.x.

Scott, D. R., and D. J. Stevenson (1984), Magma solitons, Geophysical Research Letters, 11(11), 1161-1164.

Yarushina, V. M., and Y. Y. Podladchikov (2015), (De)compaction of porous viscoelasto-plastic media: Model formulation, Journal of Geophysical Research Solid Earth, 120, doi:10.1002/2014JB011258.

Yarushina, V. M., Y. Y. Podladchikov, and J. A. D. Connolly (2015), (De)compaction of porous viscoelastoplastic media: Solitary porosity waves, Journal of Geophysical Research Solid Earth, 120, doi:10.1002/2014JB011260.